# Identification of Pathogens Causing Alfalfa Fusarium Root Rot in Inner Mongolia, China

Le Wang [1], Na Wang [2], Jialiang Yu [1], Jie Wu [1,2], Huan Liu [1], Kejian Lin [1] and Yuanyuan Zhang [1,*]

[1]  Key Laboratory of Biohazard Monitoring, Green Prevention and Control for Artificial Grassland, Ministry of Agriculture and Rural Affairs, Institute of Grassland Research of Chinese Academy of Agricultural Sciences, Hohhot 010010, China

[2]  College of Horticulture and Plant Protection, Inner Mongolia Agricultural University, Hohhot 010018, China

*  Correspondence: zhangyuanyuan01@caas.cn

**Abstract:** Alfalfa Fusarium Root Rot (AFRR) is a serious soil-borne disease with a complex pathogenicity. Diseased samples suspected of AFRR were collected from Hohhot, Ordos, Hulunbeier, Chifeng, and Bayannur in Inner Mongolia, China, leading to 317 isolates. The isolates were identified as *Fusarium acuminatum*, *F. solani*, *F. equiseti*, *F. incarnatum*, *F. oxysporum*, *F. avenaceum*, *F. verticillioides*, *F. proliferatum*, *F. falciforme*, *F. tricinctum*, *F. virguliforme*, and *F. redolens*, and the results of pathogenicity testing showed that 12 *Fusarium* species could cause alfalfa root rot. Among these, *F. verticillioides*, *F. falciforme*, and *F. virguliforme* have not previously been reported to cause AFRR in China. Although the population structure of the pathogens differed in different regions, the dominant pathogenic species was *F. acuminatum*. Fungicide toxicity tests showed that seven fungicides inhibited *F. acuminatum*, of which fludioxonil, kresoxim-methyl, and triadimefon were found to be strongly toxic towards *F. acuminatum* with $EC_{50}$ values of 0.09, 2.28, and 16.37 μg/mL, respectively, suggesting that these could be used as alternative fungicides for the control of AFRR. The results of this study can provide a theoretical basis for exploring the occurrence and epidemiology of alfalfa root rot and strategies for its control.

**Keywords:** Alfalfa Fusarium Root Rot; *Fusarium* spp.; molecular identification; fungicide

## 1. Introduction

Alfalfa (*Medicago sativa*) is a perennial herb belonging to the Leguminosae family. It is known as the 'king of forage' because of its wide adaptability, high yield, and high content of protein, vitamins, and minerals [1]. It also plays an irreplaceable role in the improvement of the ecological environment and the development of grassland animal husbandry [2]. Alfalfa is planted over about 32.2 million hm$^2$ in the world and is the fourth most widely grown crop in the USA behind only corn, soybean, and wheat [3]. In China, the alfalfa planting area is about 4 million hm$^2$, ranking first among all kinds of artificial grassland, and is mainly grown in the northwest, north, and northeast regions of the country [4]. Further development of the alfalfa industry has occurred in recent years due to the national adjustment of planting structure, and the incidence of alfalfa disease has concerned both producers and researchers [5]. Although the statistics are incomplete, it is estimated that there are about 90 diseases affecting alfalfa in China [3]. Of these, alfalfa root rot caused by *Fusarium* spp. (AFRR) results in the destruction of the roots until the entire plant dies, thus seriously affecting the yield and quality of forage [6]. The mortality rate is more than 60%, and it is estimated that as many as 92% of fields are seriously affected by the disease [3].

At present, 35 pathogens responsible for root rot pathogens have been reported, including twenty species of *Fusarium*, twelve other pathogenic fungi, two bacteria, and one nematode [7,8]. Among the fungi, *Fusarium* spp., *Rhizoctonia* spp., *Pythium* spp., *Phytophthora* spp., and *Aphanomyces* spp., amongst others, are the main pathogens causing alfalfa root rot. Additionally, the disease is usually not caused by one kind of pathogenic

fungus, but by multiple kinds of pathogenic fungi [9,10]. The population structures and dominant species of alfalfa root rot pathogens differ in different regions. Therefore, the identification of the pathogen species causing AFRR plays an important role in the effective prevention and control of the disease. The development of molecular biology has resulted in the use of genetic characterization and differences for the classification and identification of pathogenic fungi. The main gene loci used in the systematics of *Fusarium* spp. include β-tubulin, 28S rDNA, mtSSU rDNA, ITS, EF-lα, IGS, ATP, CAM, and RNA polymerase II second group (RPB2). These target genes have high species specificity and are widely used for PCR detection [11]. The ITS and EF-lα gene sequences are used, especially, for the identification of alfalfa root rot pathogens. Cong et al. used these two sequences for phylogenetic analysis of *F. tricinctum* causing alfalfa root rot in north China [12] while Yang et al. confirmed that *F. equiseti*, *F. incarnatum*, and *F. acuminatum* were the pathogens causing alfalfa root rot in Hohhot through analysis of the ITS and EF-lα gene sequences [13].

Alfalfa root rot is an important soil-borne disease that affected wide areas with significant damage. It also has a complex pathogenesis, caused by multiple pathogens, and is difficult to control. Biological control has not proved reliable, and many farmers are thus skeptical of it, preferring the more effective and economical traditional chemical methods for the prevention and control of soil-borne diseases [14]. In the past ten years, researchers have conducted a lot of research on the screening of fungicides against alfalfa root rot, and found that carbendazim, thiophanate-methyl, kufuning, fuweijue, thiophanate-methyl, thiram, fludioxonil, tebuconazole, difenoconazole, pyraclostrobin, difenoconazole, azoxystrobin, prochloraz, silazole, prochloraz, and other fungicides have an obvious inhibitory effect on alfalfa root rot [15]; In addition, seed dressing with fungicides and soil furrow spraying can also prevent the occurrence of root rot [16]. Previously, Xu et al. determined the toxicity of nine fungicides to *F. oxysporum* f. sp. *cubense* using spore germination and growth rate measurements in tissue-cultured seedlings by root irrigation [17]. They tested the effects of root irrigation in tissue-cultured seedlings and found that carbendazim and bromozazole were effective at 225 and 300 mg/L, respectively, while the effectiveness of carbendazim at 300 mg/L and 200 mg/L exceeded 57%, approximately 10–20% higher than that of carbendazim.

Inner Mongolia is the main alfalfa-producing area in China. Recent expansions in the alfalfa-planting area have led to the widespread occurrence of AFRR, seriously affecting the development of agriculture and animal husbandry in Inner Mongolia. In order to clarify the pathogen species of AFRR in the Inner Mongolia Autonomous Region, this study used a conventional tissue separation method to isolate and purify the pathogens causing alfalfa root rot in root samples, combined with morphological and molecular biology analyses to identify the isolated strains, followed by determination of their pathogenicity according to Koch's Rule. The growth rate method was used to identify effective fungicides for the dominant *Fusarium* spp. strains, and the indoor antibacterial effects of seven fungicides on a representative strain at different concentrations were used to screen the most effective fungicides. The findings can provide a theoretical basis for exploring the occurrence and epidemiology of alfalfa root rot, as well as identifying strategies for its control.

## 2. Materials and Methods

### 2.1. Isolation of Pathogens Causing AFRR

Three hundred and five diseased root samples suspected of being infected with AFRR were collected from 12 locations in five cities, namely, Hohhot, Ordos, Hulunbuir, Chifeng, and Bayannaoer, in the Inner Mongolia Autonomous Region from June to September 2021 (Table 1). All samples were packed in paper bags, dried after preliminary cleaning, and brought back to the laboratory for use.

**Table 1.** Information on the collection of alfalfa root rot samples.

| Collection Cities | Collection Locations | Sample Number | Quantity | Gathering Time | Longitude and Latitude |
|---|---|---|---|---|---|
| Ordos | Hangjin Banner | EA | 3 | 2021.06.23 | E: 107°49′09″ N: 40°47′39″ |
| Ordos | Hangjin Banner | EB | 4 | 2021.06.23 | E: 108°45′41″ N: 40°31′29″ |
| Ordos | Dalad Banner | EC | 7 | 2021.06.23 | E: 109°53′50″ N: 40°24′06″ |
| Ordos | Dalad Banner | ED | 14 | 2021.06.23 | E: 110°24′52″ N: 40°19′52″ |
| Bayannaoer | Linhe District | BW | 27 | 2021.08.15 | E: 107°31′56″ N: 40°48′55″ |
| Hohhot | Tuzuo Banner | HS | 42 | 2021.06.18 | E: 111°46′57″ N: 40°35′06″ |
| Hohhot | Helingeer County | HM | 30 | 2021.08.01 | E: 111°49′42″ N: 40°44′37″ |
| Hohhot | New District | H | 30 | 2021.08.30 | E: 111°46′56″ N: 40°54′10″ |
| Hulunbuir | Hailar District | HX | 50 | 2021.07.09 | E: 120°00′55″ N: 49°20′42″ |
| Hulunbuir | Chenbaerhu Banner | HT | 22 | 2021.07.09 | E: 120°29′27″ N: 49°33′07″ |
| Chifeng | Aohan Banner | CJ | 40 | 2021.07.11 | E: 119°47′48″ N: 42°42′53″ |
| Chifeng | Alukerqin Banner | CA | 36 | 2021.08.18 | E: 120°16′51″ N: 43°26′57″ |

The pathogens in the diseased alfalfa samples were isolated by conventional tissue separation methods. Stains on the surfaces of the samples were rinsed off with tap water, and, after natural drying, the epidermis was gently scraped with a sterile scalpel. Tissues at the border of diseased and healthy tissue were cut into fragments of approximately 5 × 5 mm, then immersed in 70% alcohol for 1 min, rinsed with sterile water 3–5 times, and dried on sterile filter paper. The sterilized fragments were transferred to Water ager (WA) plates (six pieces per plate) with sterile tweezers and were then incubated in a 25 °C incubator for three days in the dark. Mycelia without bacteria at the edge of the colonies were picked and transferred to Potato dextrose ager (PDA) plates for culture. The marginal hyphae of the colony were placed in a 2-mL centrifuge tube containing 1 mL of sterile water. The spore concentration was adjusted to $5 \times 10^3$ spores/mL and 50 μL of the spore suspension was spread evenly on the WA plates. Single colonies were picked and cultured at 25 °C on fresh PDA plates to obtain pure cultures.

### 2.2. Identification of Pathogens Causing AFRR

The colony morphology and colour of the pure cultures were recorded after six days of incubation at 25 °C on PDA plates. The isolated strains were preliminarily identified according to the 'Fungi Identification Manual' [18] and 'Fusarium' [19]. After extraction of genomic DNA from isolated strains using a fungal DNA extraction kit (TransGen Biotech Co., Ltd.), PCR amplification of the internal transcribed spacer (ITS) and translation elongation factor 1 alpha (EF-1α) and β-tubulin (TUB) gene loci was performed. The primers were ITS1 (5′-TCCTCCGCTTATTGATATGC-3′)/ITS4 (5′-GGAAGTAAAAGTCGTAACAAGG-3′) [20], and EF1 (5′-ATGGGTAAGGAAGACAAGAC-3′)/EF2 (5′-GGAAGTACCAGTGATCATGTT-3′) [21], and Bt2a (5′-GGTAACCAAATCGGTGCTGCTTTC-3′)/Bt2b (5′-ACCCTCAGTGTAGTGACCCTTGGC-3′) [22].

Each PCR reaction was performed in a 50 μL mixture containing each of the forward and reverse primers (2 μL), 2 × Taq PCR Mix (TIANGEN Biotech Co., Ltd., Beijing, China) (20 μL), genomic DNA (2 μL), and ddH$_2$O (24 μL). For both ITS and EF-1α, the cycle parameters were an initial denaturation step at 94 °C for 5 min, 35 cycles at 94 °C for 40 s, 55 °C for 40 s, 72 °C for 40 s, and final extension at 72 °C for 10 min. The annealing temperatures were 55 °C for ITS and 56 °C for EF-1α. Additionally, for TUB, the cycle protocol was an initial denaturation step of 94 °C for 5 min; 32 cycles of 94 °C for 1 min, 58 °C for 1 min, 72 °C for 1 min; final extension of 72 °C for 10 min.

The PCR products were electrophoresed on 1.2% agarose gels for 30 min, followed by evaluation and imaging with a gel imaging system to determine whether the DNA was successfully extracted. The amplification products were purified using a PCR product purification kit (Life Technologies, Carlsbad, CA, USA) and sent to Beijing Hooseen Biotechnology Co., Ltd. for sequencing. The obtained sequences were compared with the sequences of related species in GenBank by BLAST, and the phylogenetic tree was constructed by MEGA 11.0 software with the Maximum likelihood method. The bootstrap test was set to 1000 times [23,24].

### 2.3. Pathogenicity Assay

To determine whether the isolates were pathogenic to alfalfa, representative strains were randomly selected and the pathogenicity was assessed by the root dipping inoculation method [25]. After the inoculation of the test strains on PDA plates for five days, the fungus cake was picked from the edge of the colony, placed in a wheat bran medium [26], and incubated at 25 °C for 10 days. The wheat bran in the bottle was rinsed two to three times with sterile water and filtered through four layers of gauze to obtain the conidial suspension, and the concentration was adjusted to $1 \times 10^7$ spores/mL. Alfalfa seeds with full grains were selected, disinfected with 3% NaClO for 2 min, rinsed five times with sterile water, and sowed into sterilized soil in seedling trays. When the seedlings had grown 2–3 compound leaves, the seedlings from the plug seedling were gently pulled out and the root-soil was rinsed off with tap water. The seedling roots were then completely immersed in the spore filtrate, using the inoculation of sterile water as a blank control. After soaking for 30 min, the seedlings were replanted in a nutrient bowl (13 cm high and 15 cm in diameter) containing sterilized soil, with five plants per bowl. Each strain was inoculated into 15 seedlings (in three pots) and 15 uninoculated seedlings (in three pots) were used as the blank control. The nutrient bowls were returned to the artificial climate box and watered once every three days. Thirty-five days after inoculation, the incidence of root rot on all the inoculated alfalfa seedlings was investigated.

### 2.4. Sensitivity to Seven Fungicides

A dominant *Fusarium* spp. was selected for fungicide toxicity determination. A total of seven fungicide treatments were set up in the experiment (Table 2), with five concentration gradients for each treatment. Each concentration included five replicates and a plate without fungicide was used as the blank control. The inhibitory effects of each fungicide on the mycelial growth of the strain tested were determined by the plate growth rate method [27]. The concentrations for testing were first assessed in accordance with those provided in the instructions for field use. This resulted in the selection of five concentrations of each fungicide, which were then included in PDA plates (90 mm) at 45 °C to create drug-containing plates. The edges of the fungal cakes of the strain (8 mm) after five days of culture on the PDA plate were collected, and the side with hyphae facing the medium was inoculated in the center of the fungicide-containing plates. After seven days of culture in the dark at 25 °C, the colony diameters in the different fungicide-containing plates at each concentration gradient were measured by the cross method, and the rate of inhibition of mycelial growth was calculated. The inhibitory effects of the different fungicides on pathogen growth were compared and the half-maximal effective concentration value ($EC_{50}$) was determined [28].

**Table 2.** Information on the seven fungicides used in the study.

| Drug Name | Dosage Form | Manufacturer Information |
|---|---|---|
| Triadimefon | 15% WP | Sichuan Guoguang Agricultural Co., Ltd., Jianyang, Sichuan, China |
| Kresoxim-methyl | 50% WG | BASF (China) Co., Ltd., Shanghai, China |
| Mancozeb | 70% WP | Sichuan Guoguang Agricultural Co., Ltd., Jianyang, Sichuan, China |
| Fine frost · manganese zinc | 68% WG | Syngenta (China) Investment Co., Ltd., Shanghai, China |
| Ene acyl intermediate | 25% WP | Fujian Kaili Biological Products Co., Ltd., Zhangzhou, Fujian, China |
| Metalaxyl-M | 35 g/L EC | Syngenta (China) Investment Co., Ltd., Shanghai, China |
| Fludioxonil | 25 g/L FS | Syngenta Nantong Crop Protection Co., Ltd., Nantong, Jiangsu, China |

Notes: WP, wettable powder; WG, water-dispersible granules; EC, emulsifiable concentrate; FS. suspension seed coating agent.

Pure growth (cm) = colony diameter after culture (cm) − inoculated fungus cake diameter (cm).

Growth inhibition rate (%) = (control colony diameter-treated colony diameter)/control colony pure growth × 100%.

The logarithm of the concentration (X) and the percentage probability value (Y) of inhibiting colony growth was calculated. The virulence regression equation, correlation coefficient, and $EC_{50}$ value for each fungicide against the fungi were obtained by the least squares method.

## 3. Results

### 3.1. Morphological Identification of Pathogens Causing AFRR

A total of 425 isolates were obtained through the isolation and purification of diseased samples. According to the colony morphologies of strains, 317 isolates were identified as *Fusarium* spp., which were further divided into 12 groups (Figure 1) as follows. Group 1: The aerial hyphae of the strains represented by strain HX14-1 were lush, cotton-like to blanket-like, white to pink, with a camel-like color in the middle, while the back of the colony appeared wine-red. Group 2: The aerial hyphae of the strains represented by strain BW12-1 were thin linear, white to light gray, and the back of the colony was yellowish brown. Group 3: The aerial mycelia of the strains represented by strain CJ38-3 were white and filamentous at the beginning, gradually changing to a light camel-colored cotton wool appearance, while the colony was irregular in shape, and the back of the colony was light brown. Group 4: The aerial hyphae produced by the strains represented by strain HS17-2 were pale yellow velvet, with an orange depression in the middle of colonies, and the back of the colony was pale yellow. Group 5: The aerial hyphae of the strains represented by strain CA28-4 were filamentous, white to pale lotus in color, while the back of the colony produced a light purple pigment that later darkened to dark purple. Group 6: The colonies of the strains represented by strain EA2-2 were nearly round, and the mycelia were white, hairy, and vigorous, with the mycelia in the outer circle showing lush growth and the bottom producing purple pigment. Group 7: The aerial hyphae of the strains represented by strain H7 appeared as star-shaped curly wool. No pigment was produced, and the back of the colony was white. Group 8: The aerial hyphae of the strains represented by strain HM19-1-1 were woolly and spread on the medium with the back of the colony appearing light purple in color. Group 9: The aerial mycelia of the strains represented by strain HM8-1-1 were not very rich, white to pink in color, while there was light pink pigmentation on the medium. Group 10: The strains represented by strain HS14-3 grew rapidly on PDA medium, with luxuriant and dense mycelia, showing a cluster of curly hairs. Later mycelia were yellow and the back of the colony was purple. Group 11: The strains represented by strain HX22-1 grew slowly on PDA, with colonies that were usually white; the aerial hyphae were more luxuriant, and reddish-brown slime heaps were seen in the center of the colony that were white to pale yellow on the back of the colony. Group 12: The aerial mycelia of the strains represented by strain HX47-3 appeared as flocculent white cotton with lush growth, while the back of the colony was light yellow.

### 3.2. Molecular Identification of Pathogens Causing AFRR

Based on ITS, EF-1$\alpha$, and TUB gene sequences, a phylogenetic tree was constructed from 26 selected representative strains and 17 reference strains found in the database. The results showed that all strains were divided into six large groups, among which the test strains were divided into five large groups and twelve small groups (Figure 2). All groups except the peripheral strain *Lasiodiplodia theobromae* contained a variety of *Fusarium*. There were 11 strains in 'GroupI', and the strains were divided into three subgroups, indicating that *F. acuminatum*, *F. avenaceum* and *F. tricinctum* were closely related. Among them, strains BW7-1, HS5-2 and HX14-1 were clustered with *F. acuminatum*, strains EA2-2, H11-2 and HT15-1-2 were clustered with *F. avenaceum*, and strains HS14-3 were clustered with *F. tricinctum*, with branch support rates of 99%, 100%, and 99%, respectively. 'GroupII' had a total of 8 strains, which were divided into two subgroups, namely *F. equiseti* and *F. incarnatum*. Strains BW22-1, HS34-3, and CJ38-3 were clustered with *F. equiseti*, with a branch support rate of 99%. Strains HM4-1-1, BW23-2 and HS6-1 were clustered with *F. incarnatum*, with branch support of 100%. 'GroupIII' contained nine strains, which were

divided into two subgroups, *F. oxysporum* and *F. verticillioides*. Among them, the tested strains BW27-1, H22, and CA28-4 clustered together with *F. oxysporum*, and strains H7 and H15-3-1 clustered together with *F. verticillioides*, with branch support rate of 100%. The six strains were grouped in 'GroupIV', which were divided into two subgroups, *F. proliferatum* and *F. redolens*. The tested strains HM19-1-1 and HM22-2-1 clustered together with *F. proliferatum*, and strain HX47-3 clustered together with *F. redolens*, with branch support of 100%. 'GroupV' contained eight strains, which were divided into three subgroups, namely *F. solani*, *F. falciforme*, and *F. virguliforme*. The tested strains CA30-4, BW12-1, and HM8-1-2 were clustered with *F. solani*, the strains HM8-1-1 were clustered with *F. falciforme*, and HX22-1 was clustered with *F. virguliforme*, with branch support of 100%.

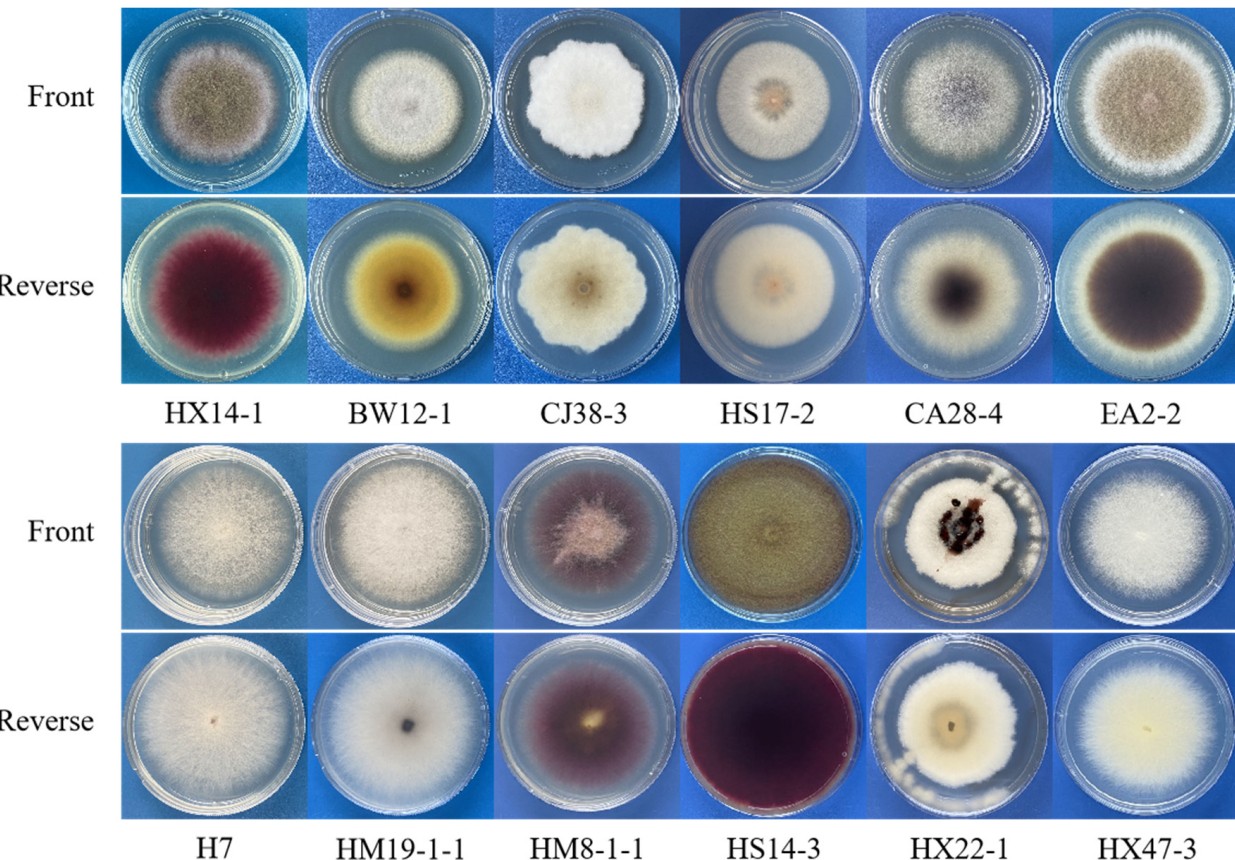

**Figure 1.** The morphological characteristics of the isolates after six days of incubation at 25 °C on PDA plates.

Based on the morphological characteristics and phylogenetic tree clustering results of the tested strains, strains BW7-1, HS5-2, and HX14-1 were identified as *F. acuminatum*, strains EA2-2, H11-2, and HT15-1-2 were identified as *F. avenaceum*, and strain HS14-3 was identified as *F. tricinctum*. The strains BW22-2, HS34-3, and CJ38-3 were identified as *F. equiseti*, the strains HM4-1-1, BW23-2, and HS6-1 were identified as *F. incarnatum*, the strains BW27-1, H22, and CA28-4 were identified as *F. oxysporum*, and the strains H7 and H15-3-1 were identified as *F. verticillioides*. Strains HM19-1-1 and HM22-2-1 were identified as *F. proliferatum*, strains HX47-3 as *F. redolens*, strains CA30-4, BW12-1, and HM8-1-2 as *F. solani*, strains HM8-1-1 as *F. falciforme*, and strain HX22-1 as *F. virguliforme*. In addition, the phylogenetic tree shows how each strain is related. Strains within the same group are closely related, such as *F. acuminatum*, *F. avenaceum*, and *F. tricinctum*. The strains under different groups were more closely related, such as *F. solani* and *F. equiseti*.

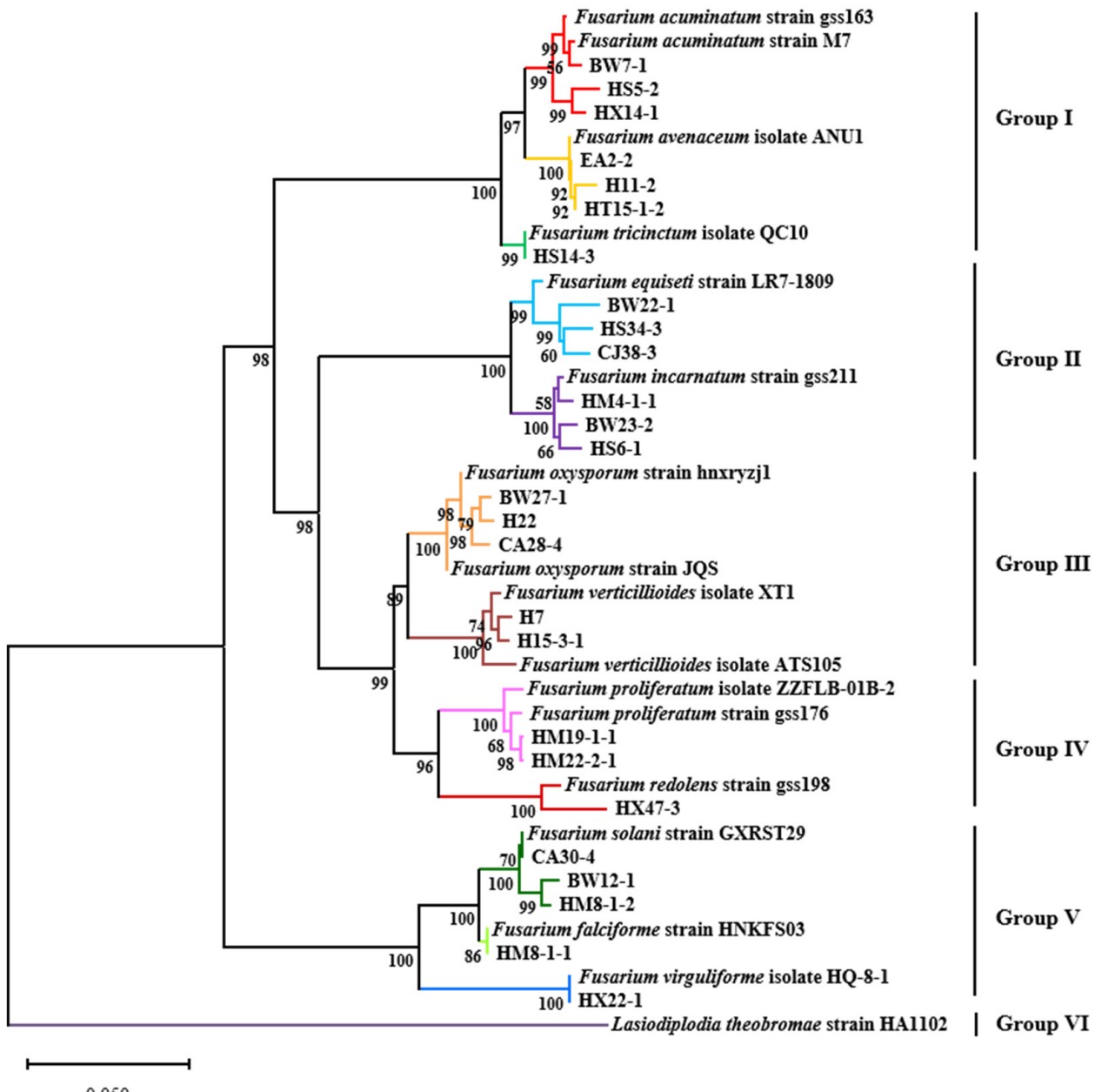

**Figure 2.** Phylogenetic tree based on the ITS+EF-1α+TUB sequences of 26 representative strains and other related *Fusarium* species retrieved from GenBank. The Maximum likelihood method was used for tree construction using Mega (version 11.0) software. Numbers at the nodes indicate bootstrap support (percent of 1000 bootstrap replicates). *L. theobromae* was used as the outgroup.

A total of 317 *Fusarium* isolates were obtained from the 12 main alfalfa planting areas in five cities of Inner Mongolia. The distribution of *Fusarium* spp. in the different urban areas is shown in Table 3. A total of 12 species of *Fusarium* were identified, namely, *F. acuminatum* (182 strains, 57.4%), *F. solani* (50 strains, 15.8%), *F. equiseti* (25 strains, 7.9%), *F. incarnatum* (23 strains, 7.3%), *F. oxysporum* (22 strains, 6.9%), *F. avenaceum* (5 strains, 1.6%), *F. verticillioides* (4 strains, 1.3%), *F. proliferatum* (2 strains, 0.6%), *F. falciforme* (1 strain, 0.3%), *F. tricinctum* (1 strain, 0.3%), *F. virguliforme* (1 strain, 0.3%), and *F. redolens* (1 strain, 0.3%).

**Table 3.** Isolation of *Fusarium* spp. strains of alfalfa root rot from different regions of Inner Mongolia, China.

| *Fusarium* Species | Collection Cities | | | | | Total |
|---|---|---|---|---|---|---|
| | Ordos | Bayannaoer | Hohhot | Hulunbuir | Chifeng | |
| *F. acuminatum* | 14 | 13 | 70 | 46 | 39 | 182 |
| *F. solani* | 2 | 7 | 7 | 16 | 18 | 50 |
| *F. equiseti* | 4 | 3 | 14 | 0 | 4 | 25 |
| *F. incarnatum* | 0 | 8 | 15 | 0 | 0 | 23 |
| *F. oxysporum* | 1 | 8 | 6 | 3 | 4 | 22 |
| *F. avenaceum* | 1 | 0 | 1 | 3 | 0 | 5 |
| *F. verticillioides* | 0 | 0 | 4 | 0 | 0 | 4 |
| *F. proliferatum* | 0 | 0 | 2 | 0 | 0 | 2 |
| *F. falciforme* | 0 | 0 | 1 | 0 | 0 | 1 |
| *F. tricinctum* | 0 | 0 | 1 | 0 | 0 | 1 |
| *F. virguliforme* | 0 | 0 | 0 | 1 | 0 | 1 |
| *F. redolens* | 0 | 0 | 0 | 1 | 0 | 1 |

The species of *Fusarium* isolated from the different locations differed. The greatest number of *Fusarium* species were isolated from Hohhot, with ten species. This was followed by Hulunbeier, Ordos, and Bayannaoer, where six, five, and five species of *Fusarium* were isolated, respectively. Chifeng had the least number of *Fusarium* species, only four. *F. acuminatum*, *F. solani*, and *F. oxysporum* were isolated from all five cities, while *F. verticillioides*, *F. proliferatum*, *F. falciforme*, and *F. tricinctum* were only isolated from Hohhot, and *F. virguliforme* and *F. redolens*, were only isolated in Hulunbeier. These results indicated that the population structure of alfalfa root rot pathogens differed across different regions. In addition, the number of different species in the same region varied greatly, and the dominant population structure of *Fusarium* spp. in each region was different. The dominant population structure of *Fusarium* species in the Ordos included *F. acuminatum*, *F. equiseti*, and *F. solani* while the population structure Bayannaoer included *F. acuminatum*, *F. incarnatum*, and *F. oxysporum* as dominant species. *F. acuminatum*, *F. incarnatum*, and *F. equiseti* were dominant in Hohhot, while the dominant *Fusarium* species in Hulunbeier were *F. acuminatum* and *F. solani*, as seen in Chifeng also. The species that was present in the largest number of isolates in each region was *F. acuminatum*. In terms of both species and the number of isolated strains, *F. acuminatum*, which accounted for 57.4%, was the most dominant species, followed by *F. solani*, *F. equiseti*, *F. incarnatum*, and *F. oxysporum*, which were widely distributed in Inner Mongolia (Table 3).

The results on the isolation and identification of the pathogens causing AFRR indicated that the disease was not only caused by infection with a single pathogen but could also be caused by multiple pathogens (Figure 3). Most of the diseased alfalfa samples (147) were infected with a single *Fusarium* species, mainly *F. acuminatum*, *F. incarnatum*, and *F. oxysporum*, while 71 diseased samples were infected with a combination of two *Fusarium* species, such as the combinations of *F. oxysporum* and *F. equiseti*, *F. acuminatum*, and *F. oxysporum*, and *F. acuminatum* and *F. incarnatum*. Moreover, there were a number (10) of diseased samples that were infected by a combination of three *Fusarium* species, such as *F. acuminatum*, *F. oxysporum*, and *F. solani*.

### 3.3. Pathogenicity

After 35 days of inoculation, all the inoculated alfalfa seedlings were severely stunted and had developed symptoms of severe chlorosis, with the branches of some plants becoming dry or the whole plant withered, whereas the control plants remained healthy (Figure 4). The same isolates were re-isolated from the roots of the inoculated plants and identified using both the morphological and molecular approaches as described above, indicating that all the *Fusarium* spp. tested were the pathogens responsible for AFRR.

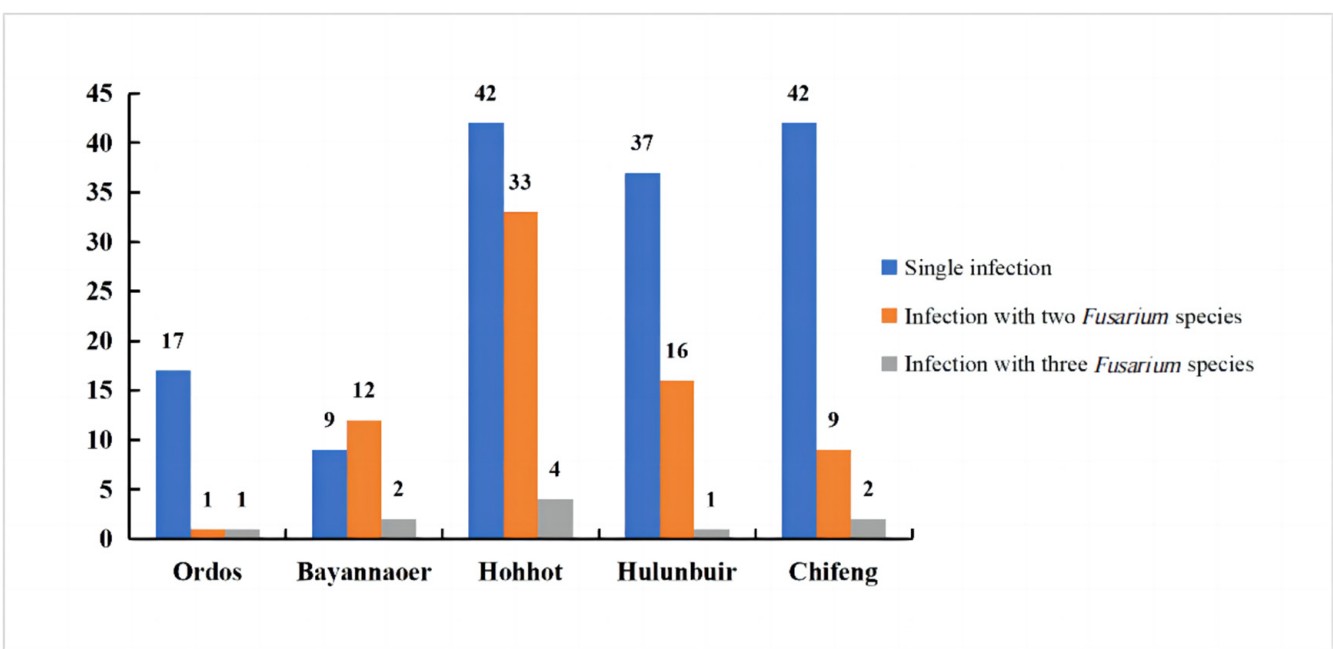

**Figure 3.** *Fusarium* infection of alfalfa in different regions of Inner Mongolia, China. The blue column indicates that only one *Fusarium* species was isolated from alfalfa samples, the orange column indicates that two *Fusarium* species were isolated from alfalfa samples, and the gray column indicates that three *Fusarium* species were isolated from alfalfa samples.

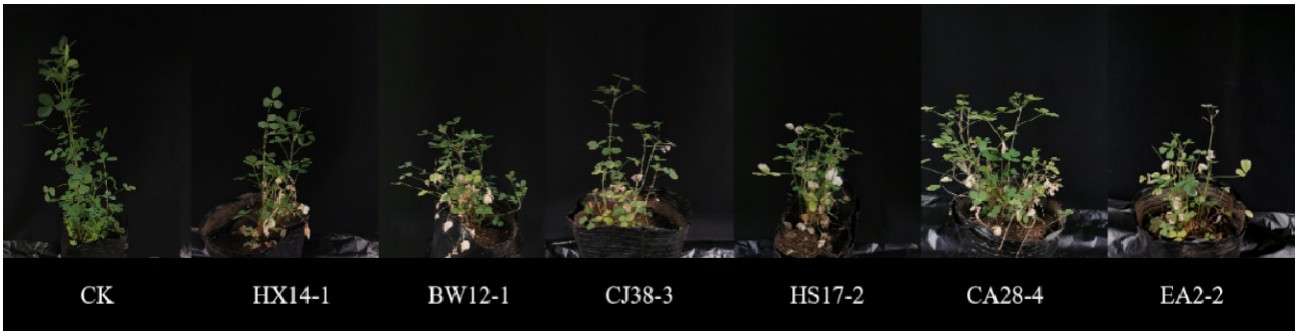

**Figure 4.** The symptoms of alfalfa seedlings 35 days after inoculation with *Fusarium* spp. strains. CK: sterile water; HX14-1: *F. acuminatum*; BW12-1: *F. solani*; CJ38-3: *F. equiseti*; HS17-2: *F. incarnatum*; CA28-4: *F. oxysporum*; EA2-2: *F. avenaceum*.

*3.4. Toxicity Determination of Seven Fungicides on Fusarium Acuminatum*

The results of the inhibitory effects of seven fungicides on the *F. acuminatum* strain showed that all seven fungicides inhibited the growth of *F. acuminatum*, and that the colony growth of *F. acuminatum* at different fungicide concentrations varied according to the dose (Table 4). The colony diameters increased as the dilution ratio of the fungicide increased, indicating that the greater the concentration of the fungicide, the better the inhibitory effect on the growth of *F. acuminatum*. High concentrations (512 µg/mL) of metalaxyl m had the best inhibitory effect on mycelia, and the inhibition rate was 71.53%. The inhibition rates of triadimefon, fine frost manganese zinc, and ene acyl intermediate were 69.17%, 64.80%, and 63.84%, respectively. The inhibitory effects of kresoxim-methyl, fludioxonil, and mancozeb were relatively weak, with inhibition rates of 58.05%, 57.91%, and 53.85%, respectively.

**Table 4.** Toxicity determinations of seven fungicides on mycelial growth of *Fusarium* on alfalfa.

| Drug Name | Treatment Concentration (μg/mL) | Concentration Logarithm (x) | Inhibition Rate % | Probability Value (Y) | Virulence Regression Equation | EC$_{50}$ (μg/mL) | R |
|---|---|---|---|---|---|---|---|
| Triadimefon | 30.00 | 1.48 | 69.17 | 5.5006 | y = 2.0593x + 2.4999 | 16.37 | 0.9938 |
| | 15.00 | 1.18 | 48.55 | 4.9636 | | | |
| | 7.50 | 0.88 | 26.73 | 4.3790 | | | |
| | 3.75 | 0.57 | 7.59 | 3.5670 | | | |
| | 1.88 | 0.27 | 2.87 | 3.0993 | | | |
| Kresoxim-methyl | 100.00 | 2.00 | 58.05 | 5.2032 | y = 0.1531x + 4.9454 | 2.28 | 0.9468 |
| | 10.00 | 1.00 | 54.57 | 5.1148 | | | |
| | 1.00 | 0.00 | 49.46 | 4.9864 | | | |
| | 0.10 | −1.00 | 44.21 | 4.8543 | | | |
| | 0.01 | −2.00 | 33.29 | 4.5682 | | | |
| Mancozeb | 40.00 | 1.60 | 53.85 | 5.0965 | y = 1.0962x + 3.2451 | 39.90 | 0.9344 |
| | 30.00 | 1.48 | 45.82 | 4.8950 | | | |
| | 20.00 | 1.30 | 31.76 | 4.5257 | | | |
| | 10.00 | 1.00 | 23.19 | 4.2675 | | | |
| | 5.00 | 0.70 | 18.53 | 4.1048 | | | |
| Fine frost · manganese zinc | 75.00 | 1.88 | 64.80 | 5.3798 | y = 1.1902x + 3.1552 | 35.48 | 0.9497 |
| | 60.00 | 1.78 | 63.53 | 5.3460 | | | |
| | 45.00 | 1.65 | 50.16 | 5.0040 | | | |
| | 30.00 | 1.48 | 48.77 | 4.9692 | | | |
| | 15.00 | 1.18 | 32.66 | 4.5506 | | | |
| Ene acyl intermediate | 1600.00 | 3.20 | 63.84 | 5.3541 | y = 1.2843x + 1.1582 | 979.49 | 0.9889 |
| | 800.00 | 2.90 | 41.87 | 4.7947 | | | |
| | 400.00 | 2.60 | 29.89 | 4.4723 | | | |
| | 200.00 | 2.30 | 18.95 | 4.1202 | | | |
| | 0.00 | 2.00 | 10.72 | 3.7583 | | | |
| Metalaxyl-M | 512.00 | 2.71 | 71.53 | 5.5689 | y = 1.2884x + 2.0665 | 189.23 | 0.9927 |
| | 28.00 | 2.11 | 36.50 | 4.6549 | | | |
| | 32.00 | 1.51 | 19.82 | 4.1520 | | | |
| | 8.00 | 0.90 | 4.19 | 3.2715 | | | |
| | 2.00 | 0.30 | 0.44 | 2.3820 | | | |
| Fludioxonil | 0.10 | −1.00 | 57.91 | 5.1997 | y = 1.5103x + 6.6113 | 0.09 | 0.9807 |
| | 0.05 | −1.30 | 30.07 | 4.4776 | | | |
| | 0.03 | −1.60 | 22.20 | 4.2346 | | | |
| | 0.01 | −1.90 | 10.80 | 3.7630 | | | |
| | 0.01 | −2.20 | 4.31 | 3.2838 | | | |

The EC$_{50}$ values showed that the tested fungicides had varying toxicities toward *F. acuminatum*, with significant differences in the inhibitory effects of the seven fungicides ($p \leq 0.05$). The toxicity of the seven fungicides to *F. acuminatum* was fludioxonil > kresoxim-methyl > triadimefon > fine frost · manganese zinc > mancozeb > metalaxyl m > ene acyl intermediate, and the EC$_{50}$ values were 0.09 μg/mL (R = 0.9807), 2.28 μg/mL (R = 0.9468), 16.37 μg/mL (R = 0.9938), 35.48 μg/mL (R = 0.9497), 39.90 μg/mL (R = 0.9344), 189.23 μg/mL (R = 0.9927), and 979.49 μg/mL (R = 0.9889), respectively. Fludioxonil, kresoxim-methyl and triadimefon thus showed strong toxicity toward *F. acuminatum*, suggesting that these could be used as alternative fungicides to control AFRR.

## 4. Discussion

Because there are many species of *Fusarium*, and the differences between the species are small, it is difficult to identify *Fusarium* spp. by morphology alone. The molecular biology method is simple to operate and has high specificity and sensitivity [29], making up for the shortcomings of traditional classification methods. At present, many researchers use the ITS and EF-1α gene sequences to differentiate between *Fusarium* species [30–32]. In this study, the ITS and EF-1α gene sequences were compared to identify the species of the isolated *Fusarium* strains, and it was found that strains HX14-1, HS14-3, HM8-1-1, HS17-2, and CJ38-3 could not be distinguished their ITS sequences while these strains were accurately identified as *F. acuminatum*, *F. tricinctum*, *F. equiseti*, *F. incarnatum*, and *F. falciforme* by the EF-1α gene sequence. The ITS sequence of many *Fusarium* species has copies of non-directional evolutionarily homologous genes, which may lead to incorrect phylogenetic analysis, so it is not possible to classify and identify *Fusarium* spp. by using

the ITS sequences alone. Therefore, it is necessary to compare multiple gene sequences when conducting a classification and phylogenetic analysis of *Fusarium* spp.

The pathogens causing AFRR are complex and new pathogens continue to be reported. There are many kinds of *Fusarium* that cause alfalfa root rot, and these species vary according to country and region, with variations in dominant *Fusarium* species also apparent. *F. avenaceum, F. arthrosporioides, F. culmorum, F. poae*, and *F. scripi* were found to cause AFRR in Alberta, Canada [33] while the main pathogens responsible for AFRR in New York in the USA were *F. avenaceum, F. oxysporum*, and *F solani*, resulting in very serious root rot [34]. In Egypt, the main AFRR pathogens were observed to be *F. oxysporum, F. semitectum, F. fusarioides, F. equiseti*, and *F. acuminatum* [35] while in China, reports on the pathogens of AFRR have mainly focused on *F. oxysporum, F. acuminatum, F. avenaceum, F. solani, F. equiseti, F. tricinctum, F. incarnatum*, and *F. proliferatum*, amongst others [36]. Furthermore, the dominant *Fusarium* species in the main alfalfa-producing areas of each province are also different. In Gansu Province, AFRR in Dingxi City was mainly caused by *F. oxysporum, F. semitectum*, and *F. acuminatum* [37], while the dominant pathogens in Wuwei City were *F. chlamydosporium, F. proliferatum*, and *F. semitectum* [38]. This is consistent with the difference in dominant population structure of *Fusarium* in different cities of Inner Mongolia.

Twelve *Fusarium* species, namely, *F. acuminatum, F. solani, F. equiseti, F. incarnatum, F. oxysporum, F. avenaceum, F. verticillioides, F. proliferatum, F. falciforme, F. tricinctum, F. virguliforme*, and *F. redolens*, are the pathogens causing AFRR. Of these, *F. verticillioides, F. falciforme*, and *F. virguliforme* have not been previously reported. This increase in the number of pathogens causing AFRR presents greater challenges to the formulation of control strategies for alfalfa root rot. In addition, alfalfa root rot is not only caused by infection with only one pathogen, but also by combinations of multiple pathogens, with the combinations of pathogens appearing to be random. A few researchers have proposed that the combination of *F. oxysporum* and *F. solani, F. oxysporum*, and *F. moniliforme* can cause alfalfa root rot [39,40]. In this study, we found that the combination of two or three *Fusarium* spp. could cause AFRR, such as the combination of *F. oxysporum* and *F. equiseti, F. acuminatum*, and *F. oxysporum, F. acuminatum*, and *F. incarnatum*, and the combination of *F. acuminatum, F. oxysporum*, and *F. solani*.

There have been a lot of studies on the biological control of alfalfa root rot. For example, *Bacillus subtilis* subsp. *spizizenii* MB29, an antagonistic bacterium isolated from alfalfa roots, can inhibit the growth of *F. semitectum, F. acuminatum, F. equiseti*, and *F. oxysporum*, and promote the growth of alfalfa [41]. However, the most important biological control is 'prevention', which plays a protective role when the disease has not yet occurred [9]. If the disease is serious, the effect of using antagonistic bacteria is not as good as chemical control. Chemical control is the most direct method. Screening effective chemical agents can control the spread of root rot pathogens. The present study found that fludioxonil and kresoxim were effective against *F. acuminatum* and could thus be used as alternative fungicides to control AFRR. This is consistent with the conclusions of other similar studies [42,43]. The inhibitory effects of metalaxyl m and ene acyl intermediate were the poorest with $EC_{50}$ values as high as 189.23 and 979.49 µg/mL, respectively. This might be because metalaxyl m and ene acyl intermediate are traditional fungicides and have been widely used for a long time, resulting in the development of resistance by *F. acuminatum* [44].

The pathogenesis of AFRR is very complex, and the pathogen species differ between different ecological environments. The pathogens are mainly distributed in the soil and can invade the root from wounds at the seedling stage or during the adult stage. Therefore, there is a high requirement for the use of fungicides. However, large fungicide doses are likely to cause environmental pollution, while small amounts cannot achieve the control effect. Therefore, it is necessary to identify effective fungicides that have both broad-spectrum actions and low toxicity to effectively control the spread of pathogens causing AFRR.

**Author Contributions:** Conceptualization, L.W. and Y.Z.; methodology, L.W.; software, J.W.; validation, N.W., J.Y. and H.L.; formal analysis, L.W.; investigation, J.Y.; resources, N.W.; data curation, N.W.; writing—original draft preparation, L.W.; writing—review and editing, K.L. and Y.Z.; visualization, L.W. and H.L.; supervision, K.L. and Y.Z.; project administration, K.L. and Y.Z.; funding acquisition, Y.Z. All authors have read and agreed to the published version of the manuscript.

**Funding:** This research was funded by the National Key Research and Development Program of China (2022YFD1401103), the Central Government Guides Local Funds for Scientific and Technological Development Program (2021ZY0040), the Inner Mongolia Natural Science Foundation (2021BS03025) and the Central Public-interest Scientific Institution Basal Research Funding (1610332022003). And The APC was funded by the National Key Research and Development Program of China (2022YFD1401103).

**Data Availability Statement:** Not applicable.

**Conflicts of Interest:** The authors declare no conflict of interest.

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
