# Peer review of "Identification of Pathogens Causing Alfalfa Fusarium Root Rot in Inner Mongolia, China"

_agronomy, doi:10.3390/agronomy13020456_

Round 1

Author Response

Dear reviewer:

Thank you very much for your comments. Based on your comment and request, we have made extensive modification on the original manuscript. The point to point responds to your comments are listed as following:

Point 1: The main problem with research design stems in the fact that the authors preliminarily group the 317 isolates based solely on colony morphology and color, and subsequently utilize just one isolate as “representative” for every downstream analysis. They do so, despite they state that “there are many species of Fusarium, and the differences between the species are small, it is difficult to identify Fusarium spp. by morphology alone” in the first few lines at the Discussion section.

 Response 1: We might have described the ‘method’ section inappropriately. Actually, we first identified 425 isolates using morphological characteristics and the comparison results of ITS sequences in GenBank. After removing 108 non-Fusarium fungi, we combined the comparison results of EF1-α gene sequences in GenBank to preliminarily group the remaining 317 Fusarium isolates, which further improving the accuracy of identification. According to your suggestion, we have added TUB gene locus in the manuscript to re-determine some strains of Fusarium fungi (for the group with more than 3 strains, we randomly selected 3 strains, and for the group with less than 3 strains, we selected all strains). The results shown that our results were consistent with those of the last time, which further shown that the identification of Fusarium species was correct.

Point 2: This is the result of a typological notion, underlying traditional phytopathological practice; but such notion does not take into account the subjacent biological complexity and variability. In short, it is completely feasible the occurrence of more than one genetic lineage within each of those twelve groups, as well as several lineages could be interspersed amongst different morphology groups. If such was the case, then the chosen colonies would not be representative at all. These possibilities need to be addressed before considering just one isolate for both genetic, pathogenicity and EC50 analyses. It is anyway highly recommended (O´ Donnell et al., 2012) to characterize a significant number of isolates from each preliminary group instead of just one colony (even if proved representative), in order to capture genetic variation, and differential pathogenicity and response to chemical control.

Response 2: We combined the comparison results of EF1-α gene sequences in GenBank to preliminarily group the 317 Fusarium isolates, and the comparison results in the NCBI database were generally consistent with the preliminary identification grouping. According to your suggestion, we have added TUB gene locus in the manuscript to re-determine some strains of Fusarium fungi (for the group with more than 3 strains, we randomly selected 3 strains, and for the group with less than 3 strains, we selected all strains).

Point 3: One other underlying assumption is that lineages evolve in a dichotomic fashion. This idea has been repeatedly questioned in recent years (Götesson et al, 2002; Kauserud et al, 2007; Swithers et al, 2012; Peris et al, 2014; Gontier, 2015; Guillin et al, 2017; Vaghefi et al, 2018, among others). This situation implies that a number of loci need to be characterized before conclusions on species identification can be drawn. Moreover, it is recommended to first carry out congruence tests and subsequent multilocus phylogenetic analyses with those loci (at least five) which show significant congruency.  In the present study, authors utilize just two loci: ITS does not univocally associate “representative colonies” and species and therefore molecular identification is solely based on the EF locus which thus result in a sort of “barcode” locus. The use of barcode loci is not sustainable when reticulated evolution is on the table, and therefore at least a few more loci (ACT, CAL, TUB, others) shall be characterized for species identification. Only a congruent, multilocus phylogeny will correctly support prevalence analysis (Lücking et al., 2020). One further benefit of multilocus analysis resides in the chance it brings to evaluate the occurrence of Horizontal Gene Transfer (HGT, Guillin et al, 2017; Vaghefi et al, 2018).

Response 3: We found that in the molecular identification of Fusarium species, most researchers chose to use ITS and EF loci for analysis, so we only focused on these two loci and identified the tested strains in combination with morphological characteristics. Now, we have added the TUB locus according to your opinion and some relevant literature, and then combined the three loci ITS, EF and TUB to establish and analyze the phylogenetic tree to more accurately evaluate the species.

Point 4: With regards to the methodology chosen for phylogenetic analysis, it is important to consider that Neighbor Joining is a fast clustering algorithm that provides an approximate idea about data structure, but it is not nowadays utilized for presenting final results; instead, Maximum likelihood and/or Bayesian approaches should be considered for building a phylogeny using sequence data, since their reliability is usually much higher (Felsenstein J. 2004).

Response 4: We have selected the Maximum Likelihood method as a new clustering algorithm to construct a phylogenetic tree for the all strains tested.

Point 5: Additionally, authors also utilize just one Genbank accession for each of the twelve Fusarium “reference species”, which once again does not take into account the likely variation within the already available sequences in public databases (they also do not indicate whether they have picked up the type specimen for each species). Variability among Genbank accessions for each species also needs to be taken into account in order to avoid misclassifications and further errors in prevalence analyses.  Again, an at least two-loci phylogeny (although insufficient) might help approximating the assignment of each isolate to each reference Genbank accessions. It is important to remark that a single gene-tree does not reflect the species history. One minor point that needs further comment is why the authors utilize different species for rooting each gene-tree.

Response 5: In order to avoid further errors in wrong classification and prevalence analysis, for each species of Fusarium, we picked up more Fusarium spp. strains with all gene loci tested identification as the reference strains searched from the GenBank database. Furthermore, according to your suggestion, we have combined three gene loci to build a phylogenetic tree.

Point 6: Even if all of the above is disregarded, prevalence results (table 3) seem to be influenced by sampling effort. Therefore, it would be pertinent to utilize some rarefaction methodology first, in order to later test for significant differences in species richness among the Inner Mongolia locations.

Response 6: Actually, 425 isolates were firstly identified by using morphological characteristics and the comparison results of ITS sequences in GenBank. For the 317 Fusarium isolates, we combined morphological characteristics, ITS and EF gene sequences to preliminarily group them. Of course, in order to avoid further errors in wrong classification and prevalence analysis, we have added TUB gene locus in the manuscript to re-determine some strains of Fusarium fungi (for the group with more than 3 strains, we randomly selected 3 strains, and for the group with less than 3 strains, we selected all strains). And then, we have combined three gene loci to build a phylogenetic tree, and the results shown that our results were consistent with those of the last time, which further shown that the identification of Fusarium species was correct.

We greatly appreciate your help and concerning improvement to this manuscript. We hope that the revised manuscript is now suitable for publication.

Thank you and best regards,

Yours sincerely,

Le Wang

Reviewer 2 Report

  1. Title of study is quite relevant.
  2. Manuscript is well drafted.
  3. Minor spelling check is required 
  4. A comparative study of chemical and biopesticide may also be done.

The main focus of study is to identify root rot pathogen of Alfalfa, AFRR (Alfalfa Fusarium Root Rot) in inner Mongolia region, Diseased samples were collected, and pathogen identification was done. Some of the commonly used fungicides were tested for pathogenicity to identify potent fungicide as control agent.

Though the topic seems original however, it is location specific, same kind of work has been done previously on other economically important plants for other locations of the world (pl see -  https://doi.org/10.1111/ppa.13333 and 10.3389/fmicb.2022.961794). The proposed work is of very basic/fundamental and suggesting fungicide as control agent doubts novelty, as these are chemical agents and harmful to environment

The study help in identification of alternative fungicide for F. acuminatum, dominant pathogen of AFRR  at inner Mongolian region. That may local farmer's in saving crop from pathogen.

Methodology followed for the proposed title is appropriate,  therefore, there is not much scope.

Though author's have stated potential harm of chemical fungicides as control agent for disease. Yet study conclude that fungicides are the potent source for control of AFRR. There are many published papers on biocontrol agents to check diseases caused by microbes.

Overall :The proposed piece of work focuses on genetic variability of fungal pathogen in Inner Mongolian city causing fusarium root rot in Alfalfa. Manuscript may be accepted for publication.

Author Response

Dear reviewer:

Thank you very much for your comments. Based on your comment and request, we have made extensive modification on the original manuscript. The point to point responds to your comments are listed as following:

Point 1: Minor spelling check is required.

 Response 1: We have checked the spelling of the whole manuscript according to your suggestion and corrected the errors.

Point 2: A comparative study of chemical and biopesticide may also be done. Though author's have stated potential harm of chemical fungicides as control agent for disease. Yet study conclude that fungicides are the potent source for control of AFRR. There are many published papers on biocontrol agents to check diseases caused by microbes.

Response 2: According to your suggestion, we have supplemented a simple comparison between biological control and chemical control in the Discussion part ‘There have been a lot of studies on the biological control of alfalfa root rot. For example, Bacillus subtilis subsp. spizizenii MB29, an antagonistic bacterium isolated from alfalfa roots, can inhibit the growth of F. semitectum, F. acuminatum, F. equiseti and F. oxysporum, and promote the growth of alfalfa. However, the most important biological control is 'prevention', which plays a protective role when the disease has not yet occurred. If the disease has erupted and is serious, the effect of using antagonistic bacteria is not as good as chemical control. Chemical control is an effective and economical method, and can quickly control the disease that is about to break out. Screening effective chemical agents can effectively control the spread of pathogens causing AFRR’.

We greatly appreciate your help and concerning improvement to this manuscript. We hope that the revised manuscript is now suitable for publication.

Thank you and best regards,

Yours sincerely,

Le Wang

Reviewer 3 Report

The aim of the paper entitled Identification of Pathogens Causing Alfalfa Fusarium Root Rot in Inner Mongolia, China is to isolate and identify pathogens causing Alfalfa Fusarium Root Rot and their suitable management by chemical pesticides. As the authors identified 3 new Fusarium spp. causing this disease, the study shall be very helpful for identifying effective fungicides with broad spectrum actions to control this disease.

However, several points need a revision before this manuscript can be accepted for publication:

Abstract:

The isolates were identified as Fusarium acuminatum, F. solani, F. equiseti, F. incarnatum, F. oxysporum, F. avenaceum, F. verticillioides, F. proliferatum, F. falciforme, F. tricinctum, F. virguliforme, and F. redolens, and the results of pathogenicity testing showed that 12 Fusarium species could cause alfalfa root rot. Among these, F. verticilli-forme, F. falciforme, and F. virguliforme have not previously been reported to cause AFRR in China

-        In your 12 identifying list of Fusarium spp., there is no F. verticilli-forme but in the next line you are reporting it as a new identified spp.

In the Discussion section, you have mentioned it again ….

Twelve Fusarium species, namely, F. acuminatum, F. solani, F. equiseti, F. incarnatum, F. oxysporum, F. avenaceum, F. verticillioides, F. proliferatum, F. falciforme, F. tricinctum, F. virguliforme, and F. redolens, are the pathogens causing AFRR. Of these, F. verticilliforme, F. falciforme, and F. virguliforme have not been previously reported.

Introduction:

Among the fungi, Fusarium spp., Rhizoctonia spp., Pythium spp., Phytophthora spp., Aphanomyces spp., amongst others are the main pathogens causing alfalfa root rot, which is usually the result of a complex infection by a variety of pathogenic fungi [9-10].

-        Writing is much confusing; it should be reframed correctly.

In the past ten years, both domestic and foreign researchers have intensively investi-gated the use of different fungicides against alfalfa root rot, namely, carbendazim, thiophanate-methyl, kufuning, fuweijue, thiophanate-methyl, thiram, fludioxonil, tebuconazole, difenoconazole, pyraclostrobin, difenoconazole, azoxystrobin, pro-chloraz, silazole, prochloraz, as well as others that have been found to be effective for controlling alfalfa root rot [15]; In addition, seed dressing with fungicides and soil furrow spraying can also prevent the occurrence of root rot [16].

-      Again, writing problem, is it Ok to write “both domestic and foreign researchers” or these worlds as well others in this para can be changed with more appropriate world like ….;

                            In the past ten years, researchers of around the world have intensively investigated the use of different fungicides against alfalfa root rot, namely, carbendazim, thiophanate-methyl, kufuning,  fuweijue, thiophanate-methyl, thiram, fludioxonil, tebuconazole, difenoconazole, pyraclostrobin, difenoconazole, azoxystrobin, pro-chloraz, silazole, prochloraz etc. that have been found to be effective for controlling alfalfa root rot [15]; In addition, seed dressing and soil furrow spraying with fungicide can also prevent the occurrence of root rot [16]. ??

Biological control has not proved reliable, and many farmers are thus skeptical of it, preferring the more effective and economical traditional chemical methods for the prevention and control of soil-borne diseases [14].

-        So, you are saying chemical control of the diseases is cheaper than biological control?

molecular biology analyses to identify the isolated strains, followed by determination of their pathogenic-ity according to Koch's Rule.

-        Which is more appropriate? Koch's Rule or Koch’s postulates?

Material and methods:

2.1 Isolation of pathogens causing AFRR

The sterilized frag-ments were transferred to WA plates (six pieces per plate)

Mycelia without bacteria at the edge of the colonies were picked and transferred to PDA plates for culture

-        What does WA mean?, must be defined the first time it is used

Table 1:  what No. is representing.?  

Collection cities

Collection locations

Sample number

No.

Gathering time

Longitude and latitude

Identification of pathogens causing AFRR:

The colony morphology and color of the pure culture were observed and recorded after six days of culture on PDA plates at 25°C.

-        Reframe the sentence, delete unnecessary worlds and please don’t include computer language to science. Check the spelling of Color.

-        recorded after six days of culture on PDA plates at 25°C OR recorded after six days of incubation at 25°C on PDA plates??

Go through the manuscript and recheck it. Specially focus on term “culture” as you have mentioned the term culture many times ex. In 2.3 Pathogenicity assay: After inoculation of the test strains on PDA plates for five days, the fungus cake was picked from the edge of the colony, placed in wheat bran medium [25], and cultured at 25°C for 10 days.  / Or it should be incubated at 25°C for 10 days.  

Figure 1. The morphological characteristics of isolates on potato dextrose agar after incubation for six days incubation at 25°C

-        Reframe it

3 Results

3.1 Morphological identification of pathogens causing AFRR

Para – 3 : total of 317 Fusarium isolates were obtained from the 12 main alfalfa planting areas in five cities of Inner Mongolia. The distribution of Fusarium spp. in the different urban areas is shown in Table 4.

-        Table no. mentioned 4 …. Is it ok ??

Para – 5 : Moreover, there were a number (10) of diseased samples that were infected by a combination of three Fusarium species, such as F. acuminatum, F. oxysporum, and F. solani.

-        Kindly elaborate how do you know that 3 different species combination of fusarium caused the disease? Have you isolated and identified these 3 species from a single diseased sample ??

Author Response

Dear reviewer:

Thank you very much for your comments. Based on your comment and request, we have made extensive modification on the original manuscript. The point to point responds to your comments are listed as following:

Point 1: Abstract: The isolates were identified as Fusarium acuminatum, F. solani, F. equiseti, F. incarnatum, F. oxysporum, F. avenaceum, F. verticillioides, F. proliferatum, F. falciforme, F. tricinctum, F. virguliforme, and F. redolens, and the results of pathogenicity testing showed that 12 Fusarium species could cause alfalfa root rot. Among these, F. verticilli-forme, F. falciforme, and F. virguliforme have not previously been reported to cause AFRR in China. In your 12 identifying list of Fusarium spp., there is no F. verticilli-forme but in the next line you are reporting it as a new identified spp. In the Discussion section, you have mentioned it again ….Twelve Fusarium species, namely, F. acuminatum, F. solani, F. equiseti, F. incarnatum, F. oxysporum, F. avenaceum, F. verticillioides, F. proliferatum, F. falciforme, F. tricinctum, F. virguliforme, and F. redolens, are the pathogens causing AFRR. Of these, F. verticilliforme, F. falciforme, and F. virguliforme have not been previously reported.

 Response 1: Here we made a mistake, the name of this specie should be F. verticillioides, not F. verticilliforme. We have corrected this error in the full text.

Point 2: Introduction: Among the fungi, Fusarium spp., Rhizoctonia spp., Pythium spp., Phytophthora spp., Aphanomyces spp., amongst others are the main pathogens causing alfalfa root rot, which is usually the result of a complex infection by a variety of pathogenic fungi [9-10]. Writing is much confusing; it should be reframed correctly.

Response 2: We have re-structured this sentence: Among the fungi, Fusarium spp., Rhizoctonia spp., Pythium spp., Phytophthora spp., Aphanomyces spp., amongst others are the main pathogens causing alfalfa root rot. Additionally, the disease is usually not caused by one kind of pathogenic fungi, but by multiple kinds of pathogenic fungi.

Point 3: In the past ten years, both domestic and foreign researchers have intensively investi-gated the use of different fungicides against alfalfa root rot, namely, carbendazim, thiophanate-methyl, kufuning, fuweijue, thiophanate-methyl, thiram, fludioxonil, tebuconazole, difenoconazole, pyraclostrobin, difenoconazole, azoxystrobin, pro-chloraz, silazole, prochloraz, as well as others that have been found to be effective for controlling alfalfa root rot [15]; In addition, seed dressing with fungicides and soil furrow spraying can also prevent the occurrence of root rot [16]. Again, writing problem, is it Ok to write “both domestic and foreign researchers” or these worlds as well others in this para can be changed with more appropriate world like ….; 

Response 3: According to your suggestion, we have revised this similar description: In the past ten years, researchers have conducted a lot of research on the screening of fungicides against alfalfa root rot, and found that carbendazim, thiophanate-methyl, kufuning, fuweijue, thiophanate-methyl, thiram, fludioxonil, tebuconazole, difenoconazole, pyraclostrobin, difenoconazole, azoxystrobin, prochloraz, silazole, prochloraz and other fungicides have obvious inhibitory effect on alfalfa root rot [15].

Point 4: Biological control has not proved reliable, and many farmers are thus skeptical of it, preferring the more effective and economical traditional chemical methods for the prevention and control of soil-borne diseases [14]. So, you are saying chemical control of the diseases is cheaper than biological control?

Response 4: Chemical control is fast and effective, and can quickly control the disease that is about to break out. In addition, modern pesticides are mostly synthetic and factory-produced chemicals, which can be supplied in large quantities with low cost, and are more convenient to use and more easily accepted by farmers. So we say chemical control is relatively effective and economical.

Point 5: molecular biology analyses to identify the isolated strains, followed by determination of their pathogenicity according to Koch's Rule. Which is more appropriate? Koch's Rule or Koch’s postulates?

Response 5: Koch 's rule, also known as Koch 's postulates or Koch 's syndrome, is the operating procedure for determining the pathogens of infectious diseases. Both ‘Koch’s Rule’ and ‘Koch’s postulates’ are OK.

Point 6: Material and methods: 2.1 Isolation of pathogens causing AFRR The sterilized fragments were transferred to WA plates (six pieces per plate) Mycelia without bacteria at the edge of the colonies were picked and transferred to PDA plates for culture. What does WA mean?  must be defined the first time it is used.

Response 6: WA is the abbreviation of ‘water agar medium’. We have added the full name in the place where we first used it, and checked and revised such problems in the full manuscript.

Point 7: Table 1: what No. is representing.?

Response 7: ‘No.’ represents the ‘Number’. In order to express more clearly, we have used the word "quantity" instead of "No.".

Point 8: Identification of pathogens causing AFRR: The colony morphology and color of the pure culture were observed and recorded after six days of culture on PDA plates at 25°C. Reframe the sentence, delete unnecessary words and please don’t include computer language to science. Check the spelling of Color. recorded after six days of culture on PDA plates at 25°C OR recorded after six days of incubation at 25°C on PDA plates?

Response 8: We have re-structured this sentence as you suggested ‘The colony morphology and colour of the pure cultures were recorded after six days of incubation at 25°C on PDA plates’.

Point 9: Go through the manuscript and recheck it. Specially focus on term “culture” as you have mentioned the term culture many times ex. In 2.3 Pathogenicity assay: After inoculation of the test strains on PDA plates for five days, the fungus cake was picked from the edge of the colony, placed in wheat bran medium [25], and cultured at 25°C for 10 days.  / Or it should be incubated at 25°C for 10 days.

Response 9: Here we replace ‘cultured’ with ‘incubated’ according to your suggestion.

Point 10: Figure 1. The morphological characteristics of isolates on potato dextrose agar after incubation for six days incubation at 25°C Reframe it.

Response 10: We have re-structured this sentence according to your comment: The morphological characteristics of the isolates after six days of incubation at 25°C on PDA plates.

Point 11: 3 Results  3.1 Morphological identification of pathogens causing AFRR

Para – 3 : total of 317 Fusarium isolates were obtained from the 12 main alfalfa planting areas in five cities of Inner Mongolia. The distribution of Fusarium spp. in the different urban areas is shown in Table 4. Table no. mentioned 4 …. Is it ok ??

Response 11: Here is a low-level error we made. We have modified it to 'The distribution of Fusarium spp. in the different urban areas is shown in Table 3'.

Point 12: Para – 5 : Moreover, there were a number (10) of diseased samples that were infected by a combination of three Fusarium species, such as F. acuminatum, F. oxysporum, and F. solani. Kindly elaborate how do you know that 3 different species combination of fusarium caused the disease? Have you isolated and identified these 3 species from a single diseased sample ??

Response 12: We identified all the strains isolated from a single alfalfa diseased sample and counted the species of the strains. Three kinds of Fusarium species were isolated from a single alfalfa diseased samples, which was considered to be the occurrence of alfalfa root rot caused by the combined infection of these three Fusarium species.

We greatly appreciate your help and concerning improvement to this manuscript. We hope that the revised manuscript is now suitable for publication.

Thank you and best regards,

Yours sincerely,

Le Wang

Round 2

Reviewer 3 Report

The authors had responded to reviewer's comments accordingly. This work is adequate and understandable. However, I think authors didn’t able to understand my question but I think manuscript should be accepted for publication

So, you are saying chemical control of the diseases is cheaper than biological control?
